

# Dramatyping: a generic algorithm for detecting reasonable temporal correlations between drug administration and lab value alterations

Axel Newe

Chair of Medical Informatics, Friedrich-Alexander University Erlangen-Nuremberg, Erlangen, Germany

## ABSTRACT

According to the World Health Organization, one of the criteria for the standardized assessment of case causality in adverse drug reactions is the temporal relationship between the intake of a drug and the occurrence of a reaction or a laboratory test abnormality. This article presents and describes an algorithm for the detection of a reasonable temporal correlation between the administration of a drug and the alteration of a laboratory value course. The algorithm is designed to process normalized lab values and is therefore universally applicable. It has a sensitivity of 0.932 for the detection of lab value courses that show changes in temporal correlation with the administration of a drug and it has a specificity of 0.967 for the detection of lab value courses that show no changes. Therefore, the algorithm is appropriate to screen the data of electronic health records and to support human experts in revealing adverse drug reactions. A reference implementation in Python programming language is available.

## INTRODUCTION

Harmful reactions to pharmaceutical products (Adverse Drug Reactions, ADRs) are a well-known cause of morbidity and mortality and are one of the common causes of death in many countries (*WHO, 2008*; *Meier et al., 2015*). Even if a drug is administered correctly (i.e., correct indication, correct dose, no contraindication etc.), it still can have unwanted side effects, of which some can be harmful (*WHO, 2008*). Therefore, the effects of a drug therapy cannot be predicted with absolute certainty: all drugs have both benefits and the potential for harm (*WHO, 2008*).

Apart from the individual fate of the affected patients, ADRs are also a large economic burden because of the resources which are required for both the diagnostics and the treatment of the symptoms as well as the diseases caused by ADRs (*Stausberg, 2014*). ADR-related costs (e.g., for hospitalization, surgery or lost productivity) exceed the cost of the medications in some countries (*WHO, 2008*). Recent figures show that the situation in different countries is roughly comparable. The average cost caused by a single ADR was calculated $3420 USD for the USA (*Hug et al., 2012*), $3681 USD for Germany (*Meier et al., 2015*) and $115 USD for India (*Rajakannan et al., 2012*). This corresponds to 42%

Corresponding author
Axel Newe, axel.newe@fau.de

(USA), 85% (Germany) and 91% (India) of the respective overall per capita expenditure on health (*WHO, 2013*).

The detection, assessment, monitoring, and prevention of drug-related adverse effects is termed pharmacovigilance (*WHO, 2008*). The general standard and still the most commonly used method of ADR detection is mandatory spontaneous reporting (*Neubert et al., 2013*). In the USA, the primary source of such reports is the FDA Adverse Event Reporting System (FAERS) database (*FDA, 2014*) of the US Food and Drug Administration (FDA). Other countries have similar systems installed (*BfArM, 2015*). This approach, however, is prone to several problems, such as under-reporting and bias (e.g., influences by media coverage) (*Neubert et al., 2013*). Up to 90% of adverse events remain undetected in hospital settings, while up to one third of hospital admissions is accompanied by ADRs (*Classen et al., 2011*). Another approach to the discovery of ADRs is the manual chart review—the gold standard in pharmacoepidemiology. It is very precise but also time-consuming and it causes high staff expenses (*Neubert et al., 2013*).

An alternative approach is the analysis of Electronic Health Records (EHRs). One example is the Sentinel Initiative (*FDA, 2008*) of the FDA, which was launched in 2008 and which aims at the development and the implementation of a system which uses existing healthcare databases (including EHRs) to assess the safety of drugs and other medical products. Although research has been conducted in this area for years, detecting ADRs in clinical records remains a challenging problem (*Liu et al., 2012*). Typical problems are inaccurate data, incomplete patient stories, data transformation and the use of narrative text instead of coded data (*Hersh et al., 2013*).

However, laboratory data have been identified as suitable parameters for the detection of ADRs (*Grönroos et al., 1995*; *Ramirez et al., 2010*; *Neubert et al., 2013*). Reference material covering the influence of drug intake on laboratory test results is available in abundance (e.g., *Young et al., 1972*; *Siest, 1980*; *Young, 2000*). One of the criteria defined by the World Health Organization (WHO) for a standardized case causality assessment of ADRs considers the temporal relationship between drug intake and the occurrence of a reaction or a laboratory test abnormality (*WHO, 2005*).

This article presents a heuristic screening algorithm for the detection of reasonable temporal correlations between drug administration and lab value alterations in appropriate data. It can be parameterized and thus be adjusted to specific use cases. A reference implementation in the Python programming language is available.

## Background and related work
### Secondary use of EHR data—dramatyping
Secondary Use, i.e., the use of data collected for other purposes is one of the most important fields of medical informatics research. A subdomain of this research is phenotyping: the identification and determination of observable characteristics of data sets, or of the underlying individuals on the basis of data that for example have been stored in EHRs. The analysis of EHR data already offers many opportunities to carry out phenotyping; however, the current methods still leave much room for improvement (*Hripcsak & Albers, 2013*).

Difficulties arise from a peculiarity of ADRs: even in the same individual, ADRs are neither reliably reproducible nor reliably detectable (*Kramer, 1981*). While patients are usually treated with a drug until the *desired* effect occurs, an adverse reaction may show up during one treatment episode while another episode stays free of ADRs. This may be because of a set of confounders that are possibly hidden from the observer. Furthermore, the imperfect quality of data found in EHRs (as mentioned above) is a factor to be considered. Therefore, the term "phenotype" is not enough in this case—instead, the term "dramatype" (introduced by *Russell & Burch, 1959*) should be chosen, because "phenotyping" refers to generally observable characteristics, whereas "dramatyping" refers to intra-individual characteristics and/or shorter observation periods.

### Interactions between drugs and lab values

One subdomain of dramatyping is the detection of Drug-Lab Value Interactions (DLVI). These interactions can be caused *in vivo* (through the drug's influence on metabolic processes or by tissue damage) or *in vitro* (by the drug's influence on chemical processes during the actual laboratory tests) (*Singh, Hebert & Gault, 1972*; *Sonntag & Scholer, 2001*).

The most obvious source of evidence for DLVI is reference literature (like *Young, 2000*), as well as relevant publications (like *Young et al., 1972*; *Siest, 1980*; *Tatonetti et al., 2011* and many others). In addition, the individual prescribing information and drug monographs provide valuable information. The latter are usually structured, but in large parts written in free text; therefore, they can only be processed indirectly by means of computers (e.g., through Natural Language Processing, NLP). Furthermore, often no direct DLVI can be derived from the drug monographs; it requires, for example, implicit medical knowledge to infer that a listed "liver injury" is related to liver-specific laboratory parameters such as Aspartate Transaminase, Alanine Transaminase or Gamma-glutamyl Transferase. This transfer work has been done and published, for example, by *Neubert et al. (2013)*.

### Detection of temporal correlations

Most approaches of mining EHR databases for drug safety signals are based on measures that quantify the degree of deviation of an observed rate of a drug-related event from a control group (*Harpaz et al., 2010*), like, e.g., the Relative Reporting Ratio (*Hauben et al., 2005*). However, in order to calculate a ratio, numerator and denominator must be known, i.e., the number of observed cases and the number of expected cases must be quantifiable. Especially the former number can be difficult to figure out since the relevant cases must be documented or at least be identifiable.

DLVIs have been proven to be suitable manifestations of such drug-related events (*Grönroos et al., 1995*; *Ramirez et al., 2010*; *Neubert et al., 2013*) and the temporal correlation between the occurrence of a lab value alteration and the preceding application of a drug is one of the criteria for a standardized case causality assessment of ADRs defined by the WHO (*WHO, 2005*) as well as by other authors (e.g., *Lucas & Colley, 1992*).

*When* exactly, however, such a temporal relationship exists (and *how* exactly it must be manifested), is not defined unequivocally—not even nearly. There are multiple degrees of freedom:

- The overall timeframe can range between immediate reactions that are obvious and late events that become evident months after drug intake (like, e.g., the thalidomide tragedy of the late 1950s/early 1960s (*Curran, 1971*)).
- The lab value course itself can show various types of changes (shifts of the mean value, deviations upwards and downwards, changes of curviness...).
- The amplitude of the change is highly various as well: some liver and heart enzymes can rise tenfold or more, while other parameters like sodium or potassium only alternate in a very narrow interval, even in pathological conditions.

Since no commonly accepted definition exists, only human expert assessments of pertinent lab value courses can be referenced as gold standard. A study covering this field of research which includes an open data corpus of such assessments made by human experts was published by (*Newe et al., 2015*). An algorithm that aims to detect generic temporal correlations between the application of drugs and corresponding lab value changes, however, has—to best knowledge—not been published before.

### Confusion of definitions

Due to the manifold settings in pharmacovigilance, literature regarding effects of drugs is characterized by a confusion of definitions. A 2012 review (*Pintor-Mármol et al., 2012*) has identified 189 associated terms used in relevant publications. For the term "Adverse Drug Event" (ADE), 15 different definitions were found, for "Adverse Drug Reaction" (ADR) there were 11. This confirms the findings of a 2005 review (*Yu, Nation & Dooley, 2005*) in which 10 (ADE) and 11 (ADR) definitions were identified, and it shows that the situation has not improved in the intermediate years.

Thus, for the rest of this article, the usual keywords or abbreviations will not be used in association with the presented algorithm. Instead, the new term "Observable Drug Event" (ODE), which—to best knowledge—has never been used before, will be used. In accordance with the criteria of the World Health Organization for a standardized case causality assessment of ADRs (*WHO, 2005*) and in analogy to the definitions in *Lucas & Colley (1992)*, an "Observable Drug Event" shall be defined as an "event or laboratory test alteration with reasonable time relationship to drug intake." This definition intentionally excludes the harmfulness or desirability of the observed events or laboratory test alterations (i.e., *desired* events or lab value alterations are included). This definition also excludes all other external factors or possible confounders and considers only the *currently observed* events (i.e., the lab value changes or any other event) and the medication intake. Furthermore, this definition intentionally includes all types of relationships, including correlations as well as causal relationships.

## METHODS

### Ethics

For the development of the algorithm described below, only previously published and freely available data has been used. Human subjects were not involved during any stage of the presented research.

| Table 1 | Formula for normalizing lab values as published in *Newe et al. (2015)*. |
|---|---|
| Formula | $LV_n = \frac{LV_a - BV_l}{BV_u - BV_l}$ |
| Parameters | $LV_n$ is the normalized lab value, $LV_a$ is the actual (absolute) lab value, $BV_l$ is the patient-specific lower border value and $BV_u$ is the patient-specific upper border value. |

## Development of the algorithm

As outlined above, the timeframe in which DLVI can be observed ranges between immediate reactions and late events that occur months after drug intake. These two extremes are out of focus of this article: immediate reactions are usually obvious and long term manifestations are hard to detect by means of EHRs. Therefore, the algorithm for detecting temporal correlations between drug administration and lab value alterations described here focuses on short term reactions, since they are well suited to be unveiled by EHR analysis (*Newe et al., 2015*).

Due to the lack of an unambigious definition of "temporal correlations," the 400 curves and their assessment by human experts published in *Newe et al. (2015)* were taken as reference and gold standard for the development of the algorithm. Since using only a minority of this limited data for the development would not be reasonable (*Hawkins, 2004*), the Ground Truth data set was not divided into training data and validation data. Instead, an external dataset was used for validation (see next section). The curves and the corresponding expert assessments were analyzed by means of feature engineering in order to inductively work out the necessary classification process.

Since a remarkable number (47 out of 400; 11.75%) of these curves was classified as "no assessment possible" by the human experts, the algorithm has been designed in a way that it produces results in a three-part nominal classification scale as well:

- "temporal correlation" if a reasonable temporal correlation between drug intake and lab value alteration is detected;
- "no change" if no correlation between drug intake and lab value alteration exists, and
- "no assessment" if the data basis does not allow for an assessment or if the incertainty is too high.

In order to avoid issues regarding the diversity of reference intervals of lab parameters (which might arise, e.g., from patient parameters like age, sex etc. or from individual laboratory equipment), the algorithm has been designed in a way that it expects normalized lab values (as proposed in formula (1) of *Newe et al., 2015*, see Table 1) as input.

The output format has also been specified to match the proposal in *Newe et al. (2015)* (Fig. 1).

Finally, some quality objectives have been specified. In (*Newe et al., 2015*), a Concordance Score $S_c$ was introduced in order to provide a simple numerical value as a means for comparing external (e.g., algorithmic) assessment results of the Ground Truth data corpus with the assessment made by human experts. The lowest $S_c$ achieved by a human expert in *Newe et al. (2015)* was 0.766 and therefore this value has been set

```xml
<?xml version="1.0" encoding="UTF-8"?>
<curve_assessment_data>
  <curve number="000">temporal correlation</curve>
  <curve number="001">no change</curve>
  <curve number="002">no assessment</curve>
  <curve number="003">no change</curve>
  <curve number="004">no change</curve>
  <curve number="005">temporal correlation</curve>
  <curve number="006">temporal correlation</curve>
  <curve number="007">no change</curve>
  <curve number="008">temporal correlation</curve>
  ...
  <curve number="391">no change</curve>
  <curve number="392">temporal correlation</curve>
  <curve number="393">temporal correlation</curve>
  <curve number="394">temporal correlation</curve>
  <curve number="395">no change</curve>
  <curve number="396">no assessment</curve>
  <curve number="397">no change</curve>
  <curve number="398">no change</curve>
  <curve number="399">no change</curve>
</curve_assessment_data>
```

**Figure 1   XML structure of the output XML.** It matches the original proposal from *Newe et al. (2015).*

as the minimum to be achieved by the algorithm. Furthermore, the worst decision that the algorithm could make was assumed to be a ''no change'' classification where actually a temporal correlation exists. Therefore, the specificity for the ''no change'' classification to be achieved by the algorithm has been specified to be 0.85 or better. Finally, since the algorithm was designed to detect temporal correlations, the sensitivity for the ''temporal correlation'' classification to be achieved by the algorithm has also been specified to be 0.80 or better. The latter two specifications, however, have not been considered mandatory.

A summary of all requirement specifications for the devlopment of the algorithm is listed in Table 2.

The reference implementation was developed using WinPython (http://winpython. github.io/) version 2.7.10.2.

## Validation of the algorithm

The validation of the algorithm was carried out in two steps in order to verify both the retrodictive value and the predictive value: first, the concordance of the algorithm

**Table 2 Requirement specifications for the development of the algorithm.**

| ID | Requirement |
|---|---|
| 1 | The algorithm *shall* classify the course of a lab value curve according to the existence of a temporal relationship between a change of this lab value curve and the administration of a drug. |
| 1.1 | The algorithm *shall* classify each lab value/medication episode into one of three categories: "temporal correlation," "no change" or "no assessment." |
| 1.2 | The algorithm *shall* use normalized lab values (according to formula (1) in *Newe et al., 2015*) as input. |
| 1.3 | The algorithm *shall* use the days of drug administration as input. |
| 2 | The algorithm *shall* write the results into an XML file as modeled in Fig. 1. |
| 3 | The Concordance Score $S_c$ of the algorithm result *shall* be larger than 0.766. |
| 4 | The algorithm *should* achieve a specifity for the "no change" classification of >0.85. |
| 5 | The algorithm *should* achieve a sensitivity for the "temporal classification" of >0.80. |

classifications with the classifications made by human experts was verified, and second, the non-existence of overfitting was verified.

While the algorithm has been developed by means of feature-engineering (i.e., induction) from the data corpus published in *Newe et al. (2015)*, the first step of the validation was carried out by means of deduction. Therefore, all curves from that data corpus were assessed by the algorithm and the classification results were then compared to the reference classifications that have been determined by human experts. By running the algorithm with varying parameters, it has been calibrated to the best possible outcome.

This result comparison was carried out by means of the DOG software application ("Data Observation Gadget") published with the data corpus article, using the result file created by the algorithm (Fig. 1) as the input and the Concordance Score $S_c$ as recommended as the output. In addition to that, a detailed breakdown of the classification results was created.

In order to ensure that the algorithm is not biased by overfitting, a validation dataset was assessed by the algorithm and the result of this assessment was compared to the result of the assessment of the Ground Truth data corpus. The criterion used for the comparison was the number of data sets that are classified by a certain step of the algorithm. The validation data consisted of the remaining 502 episodes of drug administrations with temporarily corresponding lab value observations that had not been sampled for the Ground Truth data corpus (see *Newe et al., 2015* for details).

## RESULTS

### The algorithm

The algorithm uses signal processing methods and therefore treats the lab value curve as a discrete-time biosignal. The terms which are needed to understand the following description of the algorithm are defined in Table 3.

The algorithm (Figure 2A) is generally divided into two major components: a main loop and a post-processing chain. An optional pre-processing step which checks if lab values are within the reference range [0..1] can be activated (a positive check would result in "no

**Table 3** Terms used for the description of the algorithm.

| Term | Explanation |
| --- | --- |
| Lab value curve | The normalized laboratory values of one medication episode. |
| Pre-phase | The time period before the first application of the drug. |
| Mid-phase | The time period from the first application of the drug until the last application of the drug. |
| Post-phase | The time period after the last application the drug. |
| Fitted curve | An artificially generated signal curve that has been fitted to the lab value curve (i.e., parameters like slope and intercept of the fitted curve have been optimized to match the lab value curve by means of the Levenberg–Marquardt algorithm *Moré, 1978*). |
| Low-pass filtered data | The lab value curve after application of a one-dimensional Gaussian low-pass filter (*Haddad & Akansu, 1991*) ($\sigma = 1.5$). |
| Removed outliers | The lab value curve after replacement of the two most extremely deviating values by the median. |

change"), but is deactivated by default in the reference implementation. Each step is only performed if the previous step ended without a classification result.

The main loop consists of four steps (Fig. 2B):

- Main Loop Step 1: Checks if the lab value curve is constant during the pre-phase (i.e., if the values derive less than a parameterizable tolerance from a fitted constant curve) but no longer in the mid-phase/post-phase. If this is true, it checks if the mean values of the pre-phase differ from the mean values of the mid-/post phase (i.e., if the pre-phase mean values derive more than a parameterizable tolerance from the mean values of later phases). If this is also true, this step results in "temporal correlation."
- Main Loop Step 2: Checks if the lab value curve is linear (i.e., if the values derive less than a defined parameterizable threshold from a fitted linear curve). A positive check results in "no change."
- Main Loop Step 3: Checks if the lab value curve is linear for mid-phase and post-phase (i.e., if the values derive less than a parameterizable threshold from a fitted linear curve). A positive check results in "no change."
- Main Loop Step 4: Checks if the mean values of the pre-phase differ from the mean values of the mid-/post phase (as in Main Loop Step 1). A positive check results in "temporal correlation."

This main loop is executed three times:

- with the original, unfiltered data in the first pass (Algorithm Step 1),
- with low-pass filtered data in the second pass (Algorithm Step 2), and
- with removed outliers and low-pass filtered data in the third pass (Algorithm Step 3).

The post-processing chain (Fig. 2C) comprises four additional steps:

- Algorithm Step 4: Checks low-pass filtered data for differences in slope between the phases (i.e., checks if the low-pass filtered lab values are rising or falling in the pre-phase but not in the mid-phase or the post-phase). If the pre-phase has a falling edge or rising edge, but the mid-phase and the post-phase has not, the result is "no change." If the

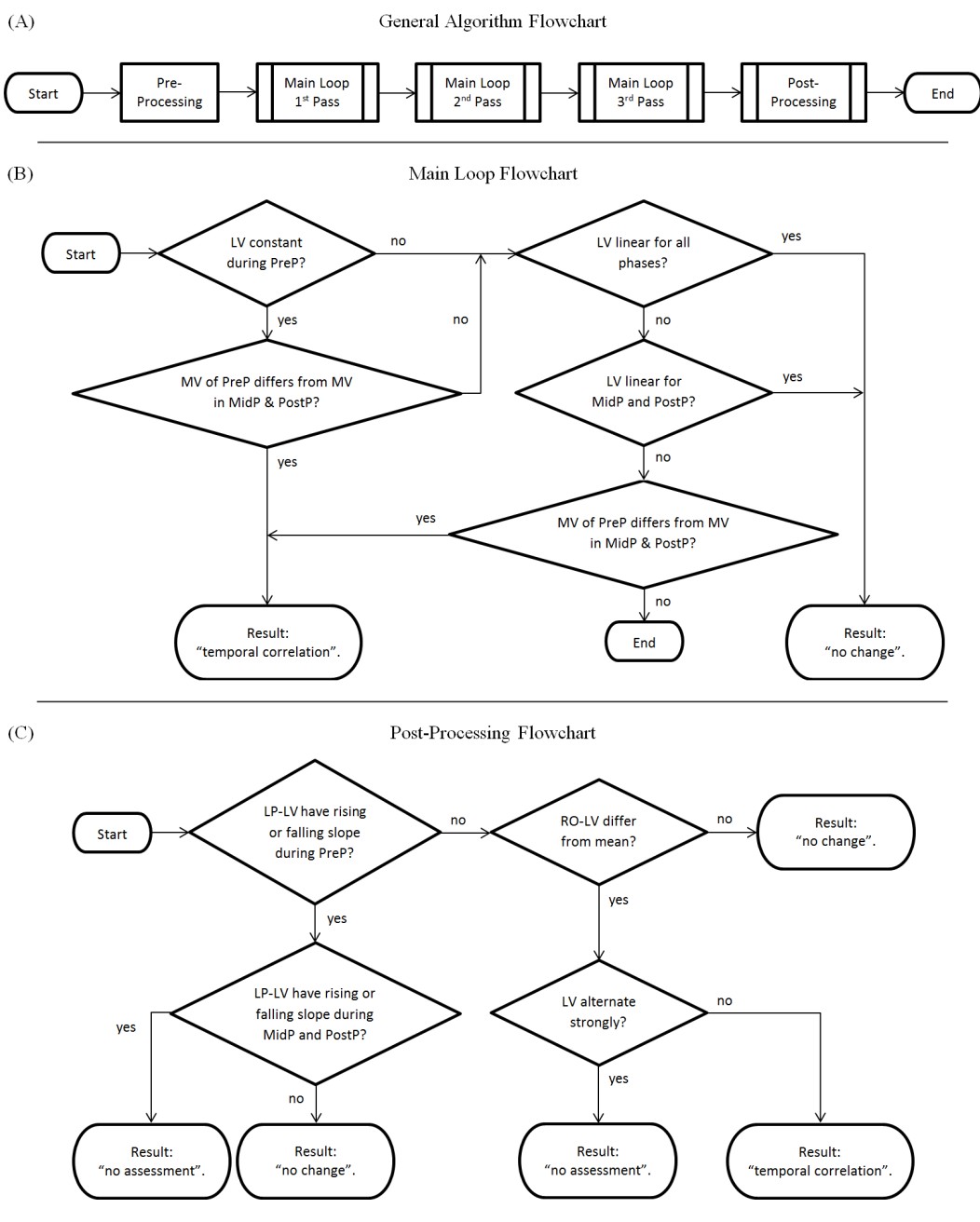

**Figure 2 Flowchart of the algorithm.** (A) Complete overview; (B) Main Loop; (C) Post-processing Chain. LV, Lab values; MV, Mean value of lab values; PreP, Pre-Phase; MidP, Mid-Phase; PostP, Post-Phase; LP-LV, Low-pass filtered lab values; RO-LV, Lab values with removed outliers.

pre-phase has a rising or falling edge, and either mid-phase or the post-phase does have a rising or falling edge as well, the result is "no assessment."

- Algorithm Step 5: Checks lab values with removed outliers for deviations from the mean value (i.e., checks if the lab values with removed outliers differ more than a parameterizable tolerance from their mean value). A negative check results in "no change."

**Table 4  Classification consensus between ground truth and algorithm output.**

| Algorithm | Ground truth | | |
|---|---|---|---|
| | No change | Temporal correlation | No assessment |
| No change | 171 | 4 | 2 |
| Temporal correlation | 40 | 124 | 37 |
| No assessment | 9 | 5 | 8 |

**Table 5  Algorithm sensitivity and specificity.**

| Classification | Sensitivity | Specificity |
|---|---|---|
| No change | 0.777 | 0.967 |
| Temporal correlation | 0.932 | 0.803 |
| No assessment | 0.170 | 0.960 |

- Algorithm Step 6: Checks if lab values are alternating strongly (i.e., if they do "zig-zag"). A positive check results in "no assessment."
- All lab value curves that could not be classified by the previous steps are finally classified as "temporal correlation."

Some of the steps seem to be redundant, but it must be kept in mind that a dataset that could be tagged with a result drops out of further processing. Therefore, the sequential processing of the steps is essential.

## Algorithm validation

As described in the Methods section, the validation of the algorithm was carried out in two steps.

First, all curves of the Ground Truth data corpus (*Newe et al., 2015*) were assessed by the algorithm and the results were then compared to the reference classifications that had been determined by human experts by means of the Concordance Score $S_c$ proposed in *Newe et al. (2015)*. The Concordance Score reached by the algorithm was $S_c = 0.803$. A more detailed breakdown of the classification results is listed in Tables 4 and 5 and a mosaic plot is available in Fig. 3.

Second, the processing results of the Ground Truth data corpus were compared in detail to the processing results of a validation dataset. For this purpose, the numbers of classifications found by each step of the algorithm were determined (Table 6 and Fig. 4). The difference between both data sets was $\Delta = 0.34 \pm 0.30\% (\Delta_{min} = 0.00\%, \Delta_{max} = 1.07\%)$. Fisher's exact test shows a significant correlation between the classification results of the Ground Truth data and of the validation data ($p = 0.9923$).

## Reference implementation

The reference implementation of the algorithm is available in Python programming language. It requires the Python packages SciPy (*Jones, Oliphant & Peterson, 2001*) version 0.16.1 and numpy (*Van der Walt, Colbert & Varoquaux, 2011*) version 1.9.3 and is available as File S1.

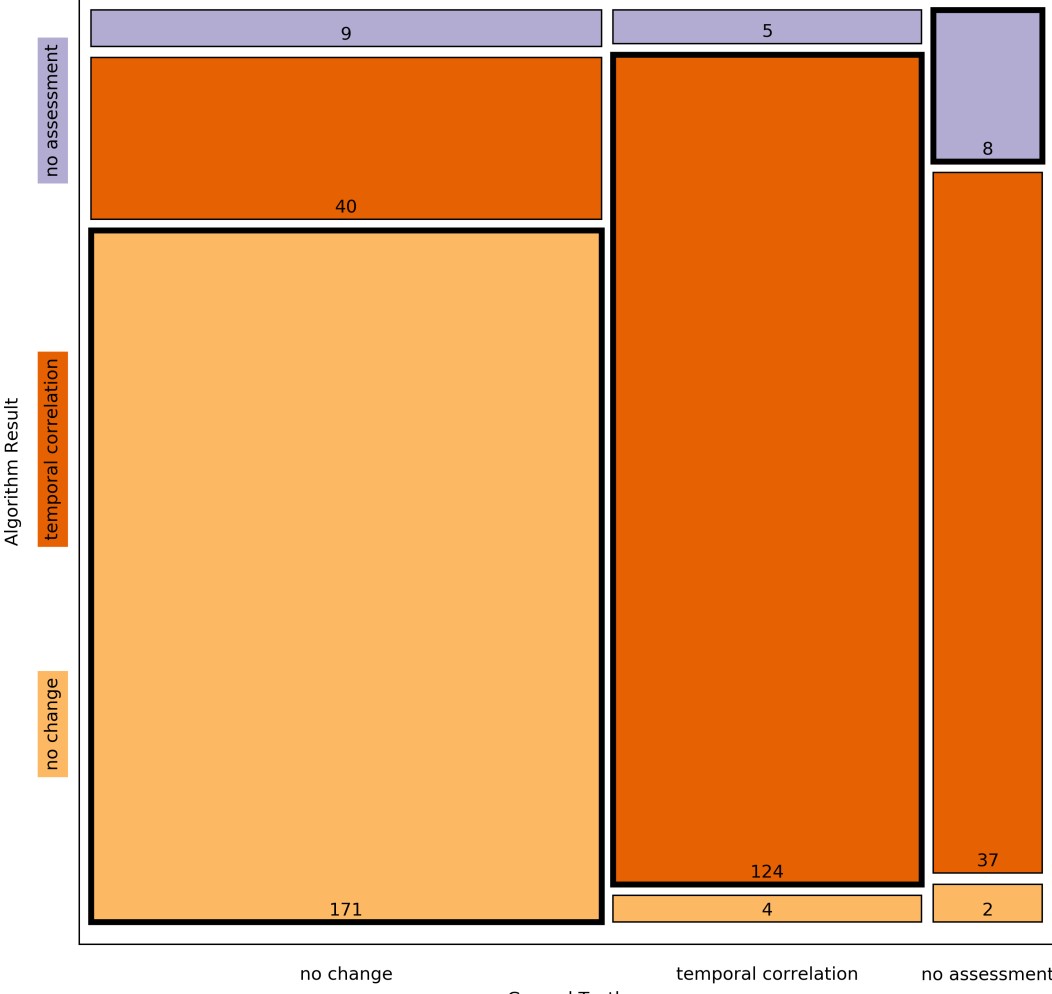

**Figure 3** **Mosaic plot of classification consensus between Ground Truth and algorithm output.** The width of the faces represents the proportion of the Ground Truth data; the height represents the proportion of the algorithm result; the figures are the absolute numbers. The boldly framed bars are those in which the algorithm achieved consensus with the ground truth.

As mentioned in the introduction, one of the problems of EHR data analyses is incomplete data. This also applies to the Ground Truth dataset: in some cases, the episodes of lab value observations are interrupted by gaps (e.g., curve #006). Since most implementations of standard functions in SciPy require continuous values, those gaps are temporarily filled with zeros during the processing, while attributing these artificial values with a very low weighting of $10^{-6}$. As a result, the error introduced by the filling of the gaps becomes negligible. When the weighting was increased for testing purposes, the assessment results started to vary at a value of $10^{-3}$; therefore $10^{-6}$ was considered to be sufficient.

The entry to the reference implementation is the TCAlgo_TestExecution.py file. In line 27, a source XML file can be specified. This source file must match the structure displayed in Fig. 5 (the name of the root element can be arbitrary).

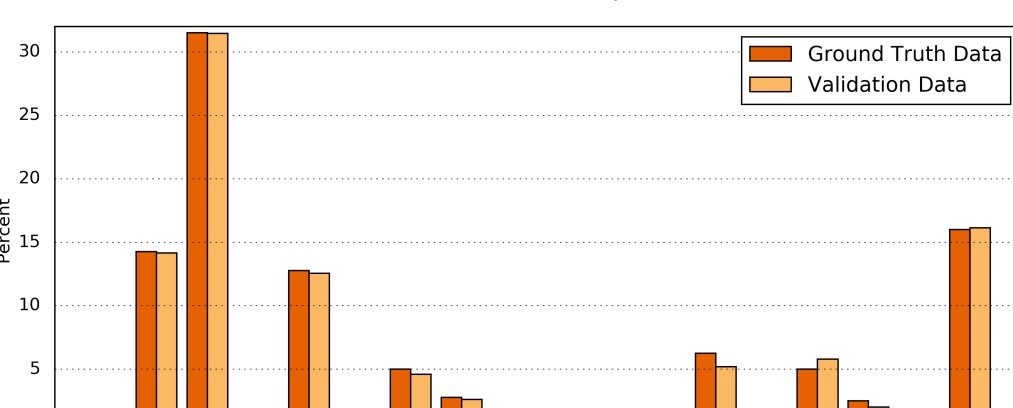

**Figure 4** **Comparison of the algorithm steps that yield an assessment result.** Plotted are the relative (%) numbers of curves in relation to the algorithm step in which they were classified for the Ground Truth data (orange) and the validation data (light orange). Note that the first step (''Preprocessing'') is disabled by default.

**Table 6** **Detailed comparison of the algorithm results for the Ground Truth data and the validation data.** This table lists the absolute (#) and the relative (%) numbers of curves in relation to the algorithm step in which they were classified. The last column shows the absolute difference of the relative numbers.

| Step Number & resulting classification | Ground Truth data | | Validation data | | abs. Δ of % |
|---|---|---|---|---|---|
| | # | % | # | % | |
| 1 — No change[a] | 0[a] | 0.00[a] | 0[a] | 0.00[a] | n/a[a] |
| 2 — Temporal correlation | 57 | 14.25 | 71 | 14.14 | 0.11 |
| 3 — No change | 126 | 31.50 | 158 | 31.47 | 0.03 |
| 4 — No change | 4 | 1.00 | 6 | 1.20 | 0.20 |
| 5 — Temporal correlation | 51 | 12.75 | 63 | 12.55 | 0.20 |
| 6 — Temporal correlation | 2 | 0.50 | 7 | 1.39 | 0.89 |
| 7 — No change | 20 | 5.00 | 23 | 4.58 | 0.42 |
| 8 — No change | 11 | 2.75 | 13 | 2.59 | 0.16 |
| 9 — Temporal correlation | 0 | 0.00 | 0 | 0.00 | 0.00 |
| 10 — Temporal correlation | 2 | 0.50 | 2 | 0.40 | 0.10 |
| 11 — No change | 0 | 0.00 | 1 | 0.20 | 0.20 |
| 12 — No change | 0 | 0.00 | 2 | 0.40 | 0.40 |
| 13 — Temporal correlation | 25 | 6.25 | 26 | 5.18 | 1.07 |
| 14 — No change | 6 | 1.50 | 9 | 1.79 | 0.29 |
| 15 — No assessment | 20 | 5.00 | 29 | 5.78 | 0.78 |
| 16 — No change | 10 | 2.50 | 10 | 1.99 | 0.51 |
| 17 — No assessment | 2 | 0.50 | 1 | 0.20 | 0.30 |
| 18 — Temporal correlation | 64 | 16.00 | 81 | 16.14 | 0.14 |

**Notes.**
[a]The first step (''Preprocessing'') is disabled by default and was not considered for the calculation of statistics.

```xml
<?xml version="1.0" encoding="UTF-8"?>
<labvalue_observations_data>
  <curve number="000">
    <medication>
      <medication_value reference_day_delta="0" />
      <medication_value reference_day_delta="1" />
      <medication_value reference_day_delta="2" />
      <medication_value reference_day_delta="3" />
      <medication_value reference_day_delta="4" />
      <medication_value reference_day_delta="5" />
      <medication_value reference_day_delta="6" />
    </medication>
    <lab_values>
      <lab_value reference_day_delta="-5" lab_value="0.000" />
      <lab_value reference_day_delta="-4" lab_value="0.100" />
      <lab_value reference_day_delta="-3" lab_value="0.200" />
      <lab_value reference_day_delta="-2" lab_value="0.300" />
      <lab_value reference_day_delta="-1" lab_value="0.400" />
      <lab_value reference_day_delta="0" lab_value="0.500" />
      <lab_value reference_day_delta="1" lab_value="0.600" />
      <lab_value reference_day_delta="2" lab_value="0.700" />
      <lab_value reference_day_delta="3" lab_value="0.600" />
      <lab_value reference_day_delta="4" lab_value="0.500" />
      <lab_value reference_day_delta="5" lab_value="0.400" />
      <lab_value reference_day_delta="6" lab_value="0.300" />
      <lab_value reference_day_delta="7" lab_value="0.200" />
      <lab_value reference_day_delta="8" lab_value="0.100" />
      <lab_value reference_day_delta="9" lab_value="0.000" />
      <lab_value reference_day_delta="10" lab_value="-0.100" />
      <lab_value reference_day_delta="11" lab_value="-0.200" />
    </lab_values>
  </curve>
  ...
  <curve number="999">...</curve>
</labvalue_observations_data>
```

**Figure 5** **XML structure of the input XML.** This XML structure is required by the importer of the reference implementation.

In line 30, the main global parameter is set: the noChangeIfInReferenceRange switch can be activated or deactivated here. If it is activated, lab values that only vary within their reference range [0..1] are automatically assessed as ''no change'' (see description of the optional pre-processing step above). This switch is deactivated by default.

The main processing function is PerformAssessmentForAllNormalizedLabValueEpisodes(). It takes all normalized lab values as input and returns a dictionary with all resulting assessments. As regards further details, please refer to the embedded documentation.

The assessment result is written to the algorithm_assesssment_result.xml file in File S1.

## DISCUSSION

### Result, purpose & usage

This article presents a heuristic, parameterizable algorithm for the detection of temporal correlations between drug administrations and lab value alterations. The default parameters

have been adjusted to match best a ground truth of such correlations published as a result of previous work (*Newe et al., 2015*). A reference implementation in Python programming language is available as File S1.

The algorithm can be used as a means for dramatyping EHR data in order to detect Observable Drug Events (ODEs) in EHRs or similar databases. It is *not* suitable (and not intended) as a sole means for the detection of ADRs, but it can serve as a module of a greater framework and thus contribute to ADR detection since ODE detection is a precursor of ADR detection. Creating this framework for ADR detection is subject of further research and therefore not within the scope of this article.

For some time, human expertise in the pharmacovigilance domain will certainly be the critical factor regarding ADR detection and ADR identification. However, assistance for these experts can well be provided by data processing tools. Instead of using it only for the identification of signals which are worthy of further investigation (as proposed in *Hauben et al. (2005)*), the algorithm presented in this article can also be used as a screening tool for the *exclusion* of datasets from further investigation. In doing so, the data that needs to be reviewed by human experts can be reduced significantly (about 44% in the case of the Ground Truth dataset) with the high specificity of 0.967 for the "no change" classification ensuring reliable results.

Especially the latter aspect should be kept in mind if the parameters are intended to be modified in future iterations or applications of the algorithm. A false-positive "no-change" classification is the worst decision since it would conceal a possible temporal correlation from the review by a human expert. Consequently, the specificity of the "no change" classification should always be kept as high as possible.

## Limitations

It is important to resist the temptation to equate detected temporal correlations with causal relationships, especially as regards ADR detection.

ADR detection is a complex task and the temporal correlation between the change of a lab value course and the admission of a drug is only one component of the criteria for a standardized case causality assessment of ADRs (*WHO, 2005*). The existence of a temporal correlation is, however, a *necessary* component. I.e., if no temporal correlation exists, the existence of an ADR can be denied.

As (*Harpaz et al., 2010*) already pointed out with respect to causality assessment, the issue of confounding is a serious concern, since it may lead to biased inference. Possible confounders are co-medication, co-morbidities, or the underlying disease itself (*WHO, 2005*; *Harpaz et al., 2010*). Since the presented algorithm processes normalized data without any information about the laboratory parameters or the drugs, it does not take into account the effect of possible confounders. However, this is intentional and a comprehensive ADR detection is not within the focus of the algorithm.

## Validation methodology and result

The objective was to design an algorithm that reproduces the assessment results of human experts whereby one has to consider that applicable data in this field is rare—in fact,
*Newe et al. (2015)* is the only available and validated source of such data. The algorithm is based on rules that were induced from this data and in order to consider as much of this limited input data as possible (*Hawkins, 2004*), the Ground Truth data set was not divided into training data and validation data. Instead, the validation was performed in two steps.

First, the rules were verified deductively in order to proof that the inductive reasoning process was valid. The default parameters yield an overall Concordance Score $S_c$ of 0.803 which is not a 100% match, but still within in the range of the Concordance Scores of the original assessors in (*Newe et al., 2015*) (minimum $S_c = 0,766$ for Assessor 05; maximum $S_c = 0,883$ for Assessor 08). Therefore, the overall result achieved by the algorithm is equal to the results of the assessments made by human experts and better than the result of the worst performing human expert.

In order to address the overfitting problem, a validation dataset with the same provenance as the Ground Truth dataset was used (*Hawkins, 2004*). The algorithms assesses both datasets with nearly the same outcome ($p = 0.9923$). Since both the Ground Truth data and the validation data originate from the same raw data it can be assumed that the algorithm produces valid results without being overfitted to the Ground Truth data set (*Hawkins, 2004*).

Another aspect that should be considered is, that one single step of the algorithm (#3) covers more than 30% of the data and that a total of four steps (#2, #3, #5, #18) covers nearly 75% of the data (in both the Ground Truth data and the validation data). The number of steps is much smaller than the number of data points in the Ground Truth dataset ($400:17 \approx 23:1$). This provides further evidence that the algorithm is not biased by overfitting.

### Other approaches of mining EHRs for drug safety signals

Several studies and large-scaled projects examined the possibilities to retrospectively detect ADRs on the basis of EHR data.

Chazard developed and implemented 236 partly very complex rules for the detection of selected ADRs that were not limited to DLVI (*Chazard, 2011*). (*Liu et al., 2012*) used a timeline-based approach to correlate drug administrations with possible ADR diagnoses that were extracted from textual medical records by means of Natural Language Processing (NLP). *Rozich, Haraden & Resar (2003)* evolved a method introduced by (*Classen et al., 1991*) and used 24 selected triggers (including 12 well-defined lab value conditions) to identify ADRs in a setting of 86 hospitals. *Harpaz et al. (2010)* published an article about the feasibility of a method that has been designed to perform an automatic mining of narrative texts in EHRs for the identification of ADE signals, and, at the same time, taking account of confounding factors.

The EU-ADR project exploited eight EHR databases of four European countries in order to detect drug safety signals (*Trifirò et al., 2011*; *Coloma et al., 2013*). The SALUS project, which has been funded with more than €3 million by the European Union (http://www.salusproject.eu/) is concerned with the creation of a proactive solution for the detection of ADRs based on EHR data. It aims to provide a standard-based interoperability framework in order to enable the performance of safety studies that can analyze real-time

patient data in communication with heterogeneous EHR systems. The Sentinel Initiative, governed by the FDA (*FDA, 2008*), is a comparable project in the United States.

The work presented in this article, by contrast, takes a more generic approach and focuses on ODEs rather than ADRs. In addition, the presented algorithm does not need any information about the involved laboratory parameters or the specific drug(s). Therefore, it can be used not only as a module of a larger framework that is aimed to detect ADRs, but also for other purposes like the assessment of data quality or the verification of desired drug effects.

## CONCLUSION

In this article, an algorithm for detecting reasonable temporal correlations between drug administration and laboratory value alterations is presented. It processes normalized values and is thus universally applicable. It has a specificity (0.967) for the detection of lab value courses that show no change in temporal correlation with drug administrations and it has a sensitivity (0.932) for the detection of lab value courses that do show a change. Therefore, it is very well suited to screen EHRs and to support human experts in the search of ADRs.

**List of Abbreviations**

| | |
|---|---|
| **ADE** | Adverse Drug Event |
| **ADR** | Adverse Drug Reaction |
| **DLVI** | Drug-Lab Value Interaction(s) |
| **EHR** | Electronic Health Record |
| **FAERS** | FDA Adverse Event Reporting System |
| **FDA** | US Food and Drug Administration |
| **NLP** | Natural Language Processing |
| **ODE** | Observable Drug Event |
| **US, USA** | United States of America |
| **WHO** | World Health Organization |
| **XML** | Extensible Markup Language |

## ACKNOWLEDGEMENTS

The present work was performed in fulfillment of the requirements for obtaining the degree "Dr. rer. biol. hum." from the Friedrich-Alexander-Universität Erlangen-Nürnberg (FAU).

### Funding
The author received no funding for this work.

### Competing Interests
The author declares that there are no competing interests.

## Author Contributions

- Axel Newe conceived and designed the experiments, performed the experiments, analyzed the data, contributed reagents/materials/analysis tools, wrote the paper, prepared figures and/or tables, reviewed drafts of the paper.

## Data Availability

The source code is provided as File S1. The used raw data (temporal_correlation_groundtruth.xml) is included in that archive. A pre-calculated result file (algorithm_assesssment_result.xml) is also included.

## Supplemental Information

Supplemental information for this article can be found online at http://dx.doi.org/10.7717/peerj.1851#supplemental-information.

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
