# Peer review of "Dramatyping: a generic algorithm for detecting reasonable temporal correlations between drug administration and lab value alterations"

_PeerJ, doi:10.7717/peerj.1851_

## Round 0.1 · original submission · Major Revisions

There are a number of shortcomings in the manuscript that were addressed by the reviewers. However, the most important shortcoming, and the reason for requiring major revisions, is that the algorithm appears to have been trained and tested on the same dataset. There are a number of alternatives to deal with this issue, for example leave-one-out or leave-many-out tests among others. The authors should make sure to train and test their algorithm in independent, statistically significant datasets prior to resubmission, at which time the publication will be re-evaluated.

·

Basic reporting

This paper proposed an algorithm that detect the correlations between drug administration and lab value.
The experiment is clearly described, and the work seems solid to me.
The dataset promises to be quite useful for similar researchers and comparative results.
Especially, the reference system also supports the clarity of this work.
However, I have several concerns with their experiments and results.
#1
Because there is no mention of the real (concrete) drag name, the discussions printed in this paper does not contain any practical information.
If possible, the author should discuss the results based on the real drug name and its effect.

Experimental design

The label consists of only three labels, temporal correlation, no change, no assessment.
In my understanding, this label set might be too rough for clinical research.
The author should add more explanation on the reason these three categories are enough.

Validity of the findings

No Comments

Additional comments

In figure 5. The number in blue box is hard to distinct. The color should be changed.

Reviewer 2 ·

Basic reporting

The authors present an algorithm for the detection of a newly defined entity called an observable drug event (ODE). This algorithm is a heuristic algorithm for "dramatying" - roughly, characterizing observable characteristics of individuals that occur over short observation periods. The authors also present a reference implementation for review.

Abstract:
The authors should report the sensitivity and specificity without comment of whether they are very high or not. If the authors would like to make such a claim they should make it in reference to another method or provide a brief explanation of why it should be considered high in the context of pharmacovigilance. Also, it would benefit the reader to see both the sensitivity and specificity for each the two categories if they are going to be reported at all.

Introduction:
- L21-22 It would be useful for the reader to distinguish between medical errors and adverse drug reactions. Yes, medical errors can be considered ADRs, but typically ADRs are considered to be adverse events when medications are used appropriately.
- L26-27 The average cost per ADR as reported by Meier et al., for example, is the overall cost where an ADR exists. This is not directly comparable to the per capita expenditures that are reported in the 2013 WHO report. The per capita cost for treating ADRs distributed across the entire population of Germany would be much less than the figure reported if it were extrapolated. Therefore, the percentages are very misleading.
- L96 Please reword the sentence including "which is such an observed-to-expected ratio". It is unclear what is meant by this statement.
- L131-141 The term "Observable Drug Event" seems very reasonable. However, the definition of "event or laboratory test alteration with reasonable time relationship to drug intake" seems somewhat broad. The definition seems to include all types of relationships, including correlations as well as causal relationships. Does the author intend to consider all drug-laboratory test pairs that have a correlation between them as "Observable Drug Events"? It seems that some aspect of causation would be desirable in such a definition. There is a note about this in the discussion, but it would be helpful to the reader to have clarification on this here as well.
- The authors do not discuss the large literature on change point analysis which seems very relevant for this work.

Experimental design

Methods:
L169 Typo. Perhaps you meant to omit "as" from this sentence.
L172 Why was the LV_n formula chosen for normalization instead of something more standard such as mean 0, standard deviation 1 normalization? This normalization scheme seems to have the same problems (susceptibility to outliers) but not many of the theoretical benefits.
L181-186 Given that S_c (Concordance Score) seems to be important for the overall message of the paper (main evaluation metric), it may be worthwhile to explain the metric fully in this paper as well.
- The majority of the results section actually describes the method (ie. the various methods used for preprocessing the data and the various "loops" involved in the overall algorithm). It would be better to restructure the paper such that the methods section contains these components.
- The most important missing element of the methods is an explanation of the process for developing the algorithm along with information on how the problem of overfitting is addressed. Was the algorithm developed without looking at any of the data? If data was involved in the development of the algorithm, is this data separate from the data that is being used to validate the algorithm? This information is critical and should be presented in the methods, not in the discussion.

Validity of the findings

Results:
- It would be helpful to understand how the developed algorithm compares to a simple baseline (e.g. a mutinomial logistic regression that uses the time series as features and predicts one of the three classes, or one of many change point analysis algorithms)
- The results for the method are positive, however, given that the authors were not blinded to the validation dataset when developing the heuristics for the algorithm, the external validity of the algorithm cannot be established. The authors claim that the effect of using the same data for development and validation is minor, but there is no evidence to support that claim.

---

## Round 0.2 · accepted · Accept

Please take a look at the minor corrections suggested by one of the reviewers and check the final manuscript for typos.

Reviewer 2 ·

Basic reporting

Consider changing the wording of the sentence, "In order to ensure that the algorithm is not biased by overfitting," to not use the term bias in this context and specify that the evaluation metric is what is being evaluated for overfitting.

Experimental design

No further comment.

Validity of the findings

Thank you for extending the study to use a validation dataset. This addresses the most major concern.

Additional comments

Minor comments:
- There is a typo on line 240 "retrodictive"